# A Female-Specific Treatment Group for ADHD—Description of the Programme and Qualitative Analysis of First Experiences

**DOI:** 10.3390/jcm13072106

**Published:** 2024-04-04

**Authors:** M. de Jong, D. S. M. R. Wynchank, M. Michielsen, A. T. F. Beekman, J. J. S. Kooij

**Affiliations:** 1Expertise Centre Adult ADHD, PsyQ, 2593 HR The Hague, The Netherlands; d.wynchank@psyq.nl (D.S.M.R.W.); m.michielsen@anteszorg.nl (M.M.); s.kooij@psyq.nl (J.J.S.K.); 2Department of Psychiatry, AmsterdamUMC/VUmc, 1081 HJ Amsterdam, The Netherlands; a.beekman@amsterdamumc.nl; 3Amsterdam Public Health Research Institute, VU Medical Centre, 1081 HV Amsterdam, The Netherlands; 4Antes Older Adults Outpatient Treatment, 3079 DZ Rotterdam, The Netherlands; 5GGZ inGeest, 1062 NP Amsterdam, The Netherlands

**Keywords:** attention-deficit/hyperactivity disorder (ADHD), women, menstrual cycle, non-pharmacological treatment, group treatment, sex hormones, premenstrual, female specific therapy, self-awareness, self-acceptance

## Abstract

**Background**: The diagnostics and treatment of attention-deficit/hyperactivity disorder (ADHD) in women remain insufficient. Fluctuations of reproductive hormones during the premenstrual period, postpartum period, and (peri)menopause are neglected, even though they impact ADHD symptoms and associated mood disorders. Therefore, we created a female-specific treatment group for women with ADHD and premenstrual worsening of ADHD and/or mood symptoms. **Methods**: We describe the group programme and underlying rationale, offering a qualitative analysis of the participants’ evaluation. **Results**: The seven bi-weekly sessions foreground the menstrual cycle and address several ADHD-specific topics in relation to this cyclical pattern. Concurrently, women track their menstrual cycle and (fluctuating) ADHD and mood symptoms with an adjusted premenstrual calendar. In total, 18 women (25–47 years) participated in three consecutive groups. We analysed the evaluation of the last group. Participants experienced the group as a safe and welcoming space. Recognition was valued by all. The topics discussed were deemed valuable, and the structure suited them well. Completing the premenstrual calendar augmented the awareness and recognition of individual cyclical symptoms. A lifespan approach increased self-understanding. Participants took their menstrual cycle more seriously, prioritising self-acceptance and self-care. **Conclusions**: Exploring a cyclical approach in a group setting seems to be a positive addition to treatment for female ADHD.

## 1. Introduction

Attention-deficit hyperactivity disorder (ADHD) in women, its characteristics, and its development across the lifespan remain insufficiently studied [1,2,3]. This is remarkable, taking into account that the sex distribution in adulthood is close to 1:1 [4,5]. As a result, treatment for women with ADHD remains suboptimal [6]. Several differences in the clinical presentation between women and men with ADHD have been described [6,7], but controversy exists [3]. It has been noted that women with ADHD when compared to men with ADHD have more mood [8,9,10], anxiety, and eating disorders [9], a higher level of emotional symptoms [11], and relational instability [12]. Emotional impulsivity and dysregulation play a substantial role in adults with ADHD [13,14,15,16]. For women with ADHD, a lack of emotional control adds greatly to their suffering [17]. For women [18], and women with ADHD especially [19,20], the menstrual cycle probably influences these symptoms.

Furthermore, it remains unknown if there are sex differences in response to treatment [21]. While psychostimulants are the mainstay of ADHD treatment, multi-modal treatment has been shown to improve treatment effects [21,22]. An expert consensus statement noted that the menstrual cycle may impact pharmacotherapeutic response but was unable to offer specific recommendations [23]. Additionally, to our knowledge, non-pharmacological treatment has not been discussed in light of hormonal fluctuations during the reproductive cycle.

Understandably, the cry for female-specific (treatment) approaches in ADHD is becoming louder [2,6,24,25,26,27,28]. Both the research and clinical fields recognise that the treatment of women is inadequate. Research has shown that the impact of sex hormones on mood and cognitive function is significant in women [18,29,30,31,32,33,34,35,36,37,38,39]. This may be more pronounced in women with ADHD [40,41,42]. Our group previously showed that women with ADHD have a higher prevalence of mood disorders than what is seen in the general population during three major periods of reproductive hormone fluctuation: the premenstrual period, postpartum, and (peri)menopause [20]. In addition, we recently described the benefits of increasing psychostimulant dosage in the premenstrual week for nine women with ADHD and premenstrual worsening of ADHD and/or mood symptoms [42]. In light of these findings, we propose that the impact of hormonal fluctuations can no longer be neglected in (treating) women with ADHD.

We, MdJ, DW and SK, are medical practitioners at a Dutch outpatient clinic (PsyQ) that specialises in treating adult ADHD and co-occurring conditions. In accordance with international guidelines [43], all patients receive a combination of pharmacological and non-pharmacological treatment, which includes psychoeducation and skills training to help them manage their symptoms. In addition, where necessary, we offer more specific emotion regulation, self-image, and schema-focused therapy groups. ADHD has high comorbidity [44,45]. On indication, we offer treatment for co-occurring sleep, depressive, anxiety, and other disorders. In line with the general consensus [23], we have been treating women and men in the same way. Currently, however, we attempt to integrate the influence of sex hormones and hormonal fluctuations into the ADHD treatment we provide [42].

Thus, deploying our professional experience, several theoretical perspectives, and recent scientific advancements, we developed a treatment group specifically for women with ADHD and self-reported premenstrual worsening of ADHD and/or mood symptoms. We foreground the effects of cyclical variations in hormones that influence mental and physical health. In addition, we integrated other relevant and possibly female-specific symptoms. We decided on a *group* programme because of the importance of social difficulties for women with ADHD [10,17]. Like others in the field [23], we recognise the importance of a lifespan approach for women with ADHD. Therefore, the second session addresses ADHD in childhood for girls and the development of symptoms and coping mechanisms over the course of a woman’s life. In sessions three, four, five, and six of the programme, respectively, we focus on emotionality, impulsivity, boundaries and safety, and triggers. Stress, its causes and consequences, and adequate management are recurring themes in our group programme.

Here, we present the full programme and underlying rationale in detail and offer a qualitative thematic analysis of the evaluation of our last group. To our knowledge, we are the first to present a therapeutic intervention for ADHD that is specifically structured around the female cyclical pattern.

## 2. Materials and Methods

We describe our group programme and the qualitative analysis of the first experiences, as offered as part of the treatment at the outpatient clinic for ADHD in adults at PsyQ, The Hague, the Netherlands. This project was carried out in accordance with the Declaration of Helsinki; safety and confidentiality were foregrounded. The Medical Ethics Review Committee of Amsterdam University Medical Centers declared that no further ethical approval was needed to conduct this study. All participants gave written informed consent after having read an information letter regarding this study.

All women had received a psychiatric assessment at our clinic, where ADHD was diagnosed or confirmed using the DIVA-5 interview [46,47]. A clinical decision was made by the respective individual practitioners to refer women to the group. All women participated in the group in addition to treatment as usual, including pharmacotherapeutic and psychological interventions aimed at minimising complaints of ADHD and co-occurring conditions in accordance with existing treatment guidelines. Women started the group at different phases of their treatment. Inclusion criteria for participation were diagnosis of ADHD, female sex, and (subjective) menstrual cycle-related worsening of ADHD and/or mood symptoms. Mild comorbidity was accepted, and exclusion criteria included severe depression, anxiety or panic disorder hindering group treatment, acute psychotic decompensation, active suicidality, and an inability to be (physically) present for more than 5 out of 7 sessions.

The design of the group and the content and structure of the sessions were based on an integration of professional experience and recent scientific advancements. First, we describe the overarching structure of the group, the session-specific content, and our rationale behind the programme. Second, we offer a qualitative analysis of the evaluation of our last group. We, MdJ and DW, have run the group three times. We refined the programme after evaluating the first and second groups: e.g., including more information about binge eating (episodes) because of the high reported prevalence amongst participants, further adjustments of the PMS calendar to capture the women’s experience more accurately, more focus on positive aspects of ADHD to counterbalance the weight of the third session, and an improved session structure.

### Data Collection and Analysis

With the consent of the participants, we audio-recorded the evaluation in the final session of the third group. This evaluation was performed in the form of a semi-structured collective interview and was conducted in Dutch. A broad open question about the participants’ experience of the group (grand tour question) and a list of relevant topics were prepared in advance. The topics were operationalised further into follow-up questions and relevant themes based on experiences with previous groups and the group programme. MdJ and DW both functioned as moderators, while DW additionally observed and recorded non-verbal communication (e.g., nodding in agreement). Directly after the evaluation, field notes were drafted together. Audio recordings were manually and verbatim transcribed by MdJ and pseudonymised. Non-verbal communication was included in the final transcript. The thematic analysis was carried out in different phases. Together, MdJ and independent researcher MM inductively coded the transcript. Code memos were kept. Iteratively, codes were clustered together until, eventually, a preliminary set of themes was generated. Together with DW, MdJ and MM iteratively refined the preliminary themes. Individually, MdJ, MM, and DW then drew up a figure that best represented the coherence of the identified preliminary themes using individual notes that had been taken. In addition, all researchers revisited the transcript for possible further adjustments. Together, MdJ, MM, and DW then discussed their figures, notes, and observations until a final consensus was reached about the (overarching) themes and their coherence. After the final consensus was reached, themes were translated from Dutch into English by MdJ and DW. Where translations posed a challenge, this was discussed until an agreement was reached. The key themes presented here, with the exception of the overarching themes, are exclusively (translations of) literal words or expressions used by the participants.

## 3. Results

We firstly describe the group programme, and secondly offer a thematic analysis of the evaluation of the last group.

### 3.1. The Group Programme

All seven sessions have a similar structure (Figure 1), last two hours, and are conducted every other week. A break of approximately one month, usually between sessions 4 and 5, is included to allow participants to implement what they have learned thus far. During the group, women track their menstrual cycle, their ADHD, mood, and somatic symptoms with an adjusted Premenstrual Dysphoric Disorder (PMDD) calendar: ‘PMS calendar’ (Appendix A). Thus, women may link any new material presented in the sessions to their own unique experiences.

#### 3.1.1. Session 1: The Menstrual Cycle

The structure of the session differs slightly, as it is the first session of the programme.

Start: Introduction of all participants and the professionals running the group (MdJ and DW). Introduction of the programme and its structure and setting ground rules. The group is designed as a safe and non-judgmental space. Amongst others, two rules are: ‘come as you are’ (e.g., inviting participants to be themselves, obviating the need to fit in socially) and ‘do what you need to do to focus’ (openness to, e.g., fidgeting, taking medication during the sessions if necessary). Inventory of additional ground rules necessary for all participating individuals to feel safe. Participants answer the following question: ‘Why do you want to participate in this group specifically?’.

Content: The impact of sex hormones on mental wellbeing [29,31,32] and brain plasticity [49]. Women with ADHD appear to have increased and more severe hormone-related complaints [20], than women without ADHD [35].

Cycle: PMS calendar is introduced and explained. From now on, this calendar will be filled in by participants between sessions.

Homework: Participants are asked to select and bring a photograph of themselves as a child, where they recognise themselves completely, unadjusted to societal expectations.

#### 3.1.2. Session 2: ADHD in Childhood for Girls

Homework: What was the experience of selecting the photograph like? Share the one chosen and explain why this particular one. What was it like to be that girl? What was the impact of social adaptation? Reflect on current relationship with the little girl in the photograph.

Content: How ADHD symptoms differ in boys and girls [7,50,51,52,53,54], aetiology of ADHD, risk of a missed diagnosis [7,52] or misdiagnosis [55] in girls and women, ADHD in puberty [56,57], ensuing coping strategies such as masking [58] or socialisation [59], and the impact hereof in later life [17,28]. Discussion of behavioural changes around the time of menarche.

Cycle: Address additional questions or problems concerning the implementation of the PMS calendar.

Homework: Full attention paid to the PMS calendar. The next session completes one month of tracking symptoms. Reflect on and discuss any apparent patterns.

#### 3.1.3. Session 3: ADHD and Emotionality

Homework: Share and discuss the implementation of the PMS calendar and discern any patterns emerging.

Content/Cycle: Menstrual cycle and associated hormone fluctuations in more detail. Premenstrual syndrome (PMS) [60] and PMDD [61] in more detail. The interrelationship between fluctuating sex hormones and ADHD (symptoms) [30,62] and the relation between oestrogen and dopamine [63]. Effect of oestrogen on the prefrontal cortex [34]. ADHD is associated with more frequent and severe premenstrual- and postnatal mood symptoms, (peri-)menopausal complaints [20] and dysmenorrhea [64]. Emotional impulsivity and self-(dys)regulation presented as possible core characteristics of ADHD [13]. Reflect on the influence of the menstrual cycle on these symptoms. End with positive sides of ADHD [65].

Homework: Reflect on and present three positive aspects of participants’ own ADHD.

#### 3.1.4. Session 4: ADHD and Impulsivity

Homework: Share and discuss personal experiences of positive aspects of ADHD. 

Content: Impulsivity can manifest in different ways [15]. In adult ADHD, it is associated with compulsive sexual behaviour, ‘sensation seeking,’ difficulties in delayed gratification, and emotional dysregulation [16]. Mutually sustainable relationship of impulsivity, restlessness, and stress. Hormonal fluctuations impact impulsive [30,62] and addictive behaviours [66]. ADHD is associated with disordered eating behaviour [67,68,69,70,71,72,73]. Eating behaviour fluctuates cyclically [74], and oestrogen influences appetite [75]. Individual experiences with fluctuating impulsivity, possible consequences and sharing of tips and tricks used to alleviate symptoms.

Homework: Plan and execute (at least) three self-care activities. These are novel activities or activities that are rarely experienced but are usually enjoyed.

#### 3.1.5. Session 5: ADHD, Boundaries, and Safety

Homework: Share and discuss self-care activities and the experience of foregrounding these.

Content: Unsafety and feeling unsafe. Sensory hyper- and hyposensitivity in women with ADHD [76], and the spectrum of consequences; any resulting ‘sensation seeking’ behaviour can lead to unsafe situations, while too many light/sound/tactile stimuli can produce overstimulation and cause feelings of unsafety. Fluctuating impulsivity and emotional dysregulation in relation to maintaining personal boundaries. Cyclically fluctuating boundaries and the role of interoception [77,78]. Prolonged masking behaviours and negative previous/current social experiences [17,79] as sources of feeling unsafe. Negative consequences of insufficient self-care. The importance of a sense of control or agency for women with ADHD [17]. Individual experiences with unsafety and feeling unsafe in relation to the theory.

Homework: Explore individual boundaries by reflecting on the following questions: (1) Where do you have adequate boundaries? (2) Where do you have inadequate boundaries? (3) When was the last time you transgressed your own boundaries? (4) Which signals does your body give you when you (almost) transgress your boundaries?

#### 3.1.6. Session 6: ADHD and Triggers

Homework: Share and discuss the answers to the questions above. What was the experience of reflecting on them like? How does the menstrual cycle influence the (experienced) boundaries?

Content: Prevalence of delayed circadian rhythm in ADHD [80] and effect of psychostimulants on late sleep [81]. How women with ADHD tackle stress and experience impairment [10] in social functioning and with time perception. Influence of premenstrual hormonal fluctuations on sleep [82], social functioning [83], emotional stress and psychological triggers [84]. Explore sensitivity to (physical/psychological) triggers in relation to the menstrual cycle (e.g., sensory stimuli, experiencing rejection, judgement, and sleep deprivation). Discuss individual experiences with triggers and their shared characteristics.

Homework: Create a plan of how to integrate what was learned in the group, a cyclical plan that considers one’s particular wants and needs. Get creative!

#### 3.1.7. Session 7: The Future

Homework: Presentation of the future plans of all participants.

Ending: Evaluation of the group programme.

Following the group, participants have the possibility to discuss and implement their individual plans for the future with their individual therapist.

### 3.2. Experiences

Six women (29–46 years) commenced the group treatment (Table 1). All completed the programme. Five were present at the final evaluation; the sixth group member was unable to attend but had sent a detailed email describing her experiences and future plans, which we shared during the final session. The recorded evaluation lasted 36 min. We, MdJ and DW, experienced the evaluation as somewhat different from the regular sessions. Especially at the beginning of the evaluation, the participants took great care to take turns speaking. After a lengthy discussion, MdJ and DW concluded that the evaluation nonetheless reflected the typical group dynamic.

All participants had received their ADHD diagnosis in adulthood. The mean time since the ADHD diagnosis was 32 months; the mean time since the commencement of treatment for ADHD at PsyQ was 10 months. At the start of the group, four of the six women used ADHD medication, of which two were on stable doses. At the final meeting, five of the six women were using ADHD medication, of which three were on stable doses (Table 1). During the programme, one participant had started an SSRI. No hormonal contraceptives were used by 4/6 participants. The combined oral contraceptive pill was used by one participant (1/6), as was a hormonal intrauterine device (1/6). For co-occurring conditions, see Box 1.

Unlike the description of the standard group above, for this particular group, we were forced to cancel the fourth session as a result of dangerous weather conditions. In consultation with the participants, we integrated the fourth into the fifth session.

Box 1Classifications based on DSM-5. *Main diagnosis*: ADHD (67%), ADD (33%) Co-occurring conditions: *Affective*: Depr-recur-mild (17%), PDD (17%), Depr-recur-part-remis (17%) *Anxiety*: GAD (33%), UAD (17%) *Trauma*: PTSD (17%), UTSD (17%)  *Other*: DSPD (33%), ASD (17%), CUD (17%)  *Somatic*: Hypertension (17%), Migraines (17%) Prevalence of diagnoses in the last group (*n* = 6). Abbreviations: ADHD: attention-deficit/hyperactivity disorder–combined subtype; ADD: attention-deficit/hyperactivity disorder–inattentive subtype; ASD: autism spectrum disorder; CUD: cannabis use disorder; Depr-recur-part-remis/mild: depressive disorder–recurring–in partial remission/mild; DSPD: delayed sleep phase disorder; GAD: generalised anxiety disorder; PDD: persistent depressive disorder (dysthymia); PTSD: post-traumatic stress disorder; UAD: unspecified anxiety disorder; UTSD: unspecified trauma and stressor-related disorder.

#### Evaluation

In our qualitative analysis of the final evaluation, we identified six key themes: safety-sharing-welcome; recognition; good fit; eyeopener; take (n) seriously; and empathy. The two overarching themes were understanding and connection (Figure 2).

Participants emphasised that they valued feeling **safe** in the group, at ease and **welcome**. They underlined the importance of **sharing** openly, honestly and with emotion. They shared previous stories and experiences as well as their current experience with the group.

I knew about the [menstrual] cycle, I have of course covered it in my studies, because I’ve studied healthcare, but it was never discussed with me in *this* way, really with emotions and stuff like that, and (we) really share it with each other (emphasis in transcript).(I2)

Last week for instance, I was very emotional, but I was actually welcomed here with open arms by everyone and I could share my story… I was not exaggerating, as all of you would actually have the same feelings [if you were in my position], so for me that recognition is great.(I3)

The participants mentioned the value of **recognition**. Some valued the group’s emphasis on the menstrual cycle, while others considered it a bit too strong. They recognised themselves in the other group members and mentioned that the structure of the topics was a **good fit** for them.

I appreciated it that for instance … sometimes emotions can invade like a kind of fireworks, that others were able to recognise it too, which made me think … ‘oh, [this is] apparently not something special, because … other people experience it too’, I liked that a lot too.(I4)

Recognition is very valuable, just like you already said, because it is something completely different to … encounter an Instagram-account online … in which you do recognise yourself, but then you just continue scrolling. Here, you are actually sitting around the table [together].(I5)

Participants mentioned multiple **eye-openers**. Several considered the childhood theme impactful, as it changed their self-regard and made them want to change some things in their lives. They found it insightful to discuss other life phases and address them from the perspective of ADHD and hormones.

Yes, and that it does not only influence your [menstrual] cycle but also other phases in your life, during puberty and later menopause … or after giving birth … and that this is once again so connected with ADHD. I also did not know this, but it again explains a lot. So if I go back in time … why was *I* such a terrible adolescent? … [What] was happening in my head? I also like it that we have discussed that.(I3)

The PMS calendar (Appendix A) was considered useful as participants were not aware that all the listed symptoms might relate to their menstrual cycle.

Yes … that there was a ready-made list of everything … you could suffer from [premenstrually] and that made you start to think, ‘oh, that is apparently something that could be related to my [menstrual] cycle’.(I1)

Premenstrual symptoms were similar between participants but not identical.

For me that was a real issue, that you struggle to get started … I noticed a big difference there … as I approached my menstruation.(I1)

For me it was itchiness. I was not at all aware of this, but now that I know it I think ‘oh right, that is something that I am a little extra sensitive to’.(I5)

For me it was feeling a bit under the weather. Yes, I feel a bit unwell when I have PMS.(I2)

Self-criticism was prevalent, while simultaneously, participants had **empathy** for other group members in similar situations. This contrast provided insight, too. At the final evaluation, participants reported that they **take** themselves and the impact of their menstrual cycle **more seriously**. They give themselves more space in the premenstrual phase, are kinder to themselves, listen more to their own needs and say ‘no’ more often. Participants mentioned improved self-acceptance. They also noted that they are more aware of possible (future) treatment options and what they might need for themselves. In addition, participants expressed that they have gained confidence in communicating about their menstrual cycle to others and can do so more clearly. These changes were described as an ongoing process.

[Having] some more clarity for myself, allows me to be more clear with other people.(I5)

There was ambivalence about the time investment required for participation in the group. Some participants expressed difficulties fitting the group in alongside work or other (treatment) appointments. While they had found the experience valuable, they were also relieved to have one thing less to fit into their calendar. Others would have preferred the programme to continue for longer. While time investment was a challenge for some, it also showed them that taking time for themselves is possible, even with busy schedules. Potential improvements to the group were suggested. Participants would have liked more time for reflection during the sessions and suggested additional relevant topics: more practical tips, explicitly addressing the costs of ADHD, and more focus on rest and relaxation. Ultimately, all agreed that a part two of the group would be desirable.

*Connection* was considered an overarching theme. We understand this as a connection with oneself, with others in the group and their similar experiences, and between experience and theory. It additionally captures the experienced value of recognising the connection between ADHD and/or mood symptoms and hormonal fluctuations during different life phases. *Understanding* also emerged as an overarching theme. Participants emphasised the importance of understanding their experiences, making sense of their past, normalising their difficulties, and developing empathy for themselves and each other.

## 4. Discussion

We describe here a group programme specifically designed for women with ADHD and self-reported premenstrual worsening of ADHD and/or mood symptoms that was offered as an adjunct to the outpatient adult ADHD treatment at PsyQ, The Hague, the Netherlands. This is the first time, to our knowledge, that a therapeutic intervention specifically tailored to the menstrual cycle in ADHD is described. Also, we present a thematic qualitative analysis of the final evaluation of our last group, which reflects the participants’ experience of the group programme. Participants considered the topics valuable and the programme structure well tailored to them. The group was experienced as a safe and welcoming space. Recognition in each other as well as in the provided theory and PMS calendar (Appendix A) was insightful and valued by all. Discussing ADHD throughout the female lifespan increased self-understanding and resulted in a shift in self-regard. At the final evaluation, participants took their menstrual cycle more seriously, prioritising self-acceptance and self-care in the luteal phase of their cycle. This was described as an ongoing process. There was ambivalence about the time investment required, and some participants found the emphasis on the menstrual cycle a bit too strong. Others valued this approach. The key themes that were identified via thematic qualitative analysis of the final evaluation were eyeopener, good fit, recognition, take(n) seriously, empathy, and safety/sharing/welcome (Figure 2). Additionally, we identified two overarching themes: *understanding* and *connection* (Figure 2).

We developed this female-specific treatment option in response to the dire need expressed by many [1,17,19,23,24,25,26,28,41,55,85], including patients. The programme is explicitly a product of combining and integrating our clinical experience with the available scientific evidence. Importantly, we do not claim that our programme design is optimal. We hope to boost the parallel and collaborative development of female-specific therapeutic options for ADHD and emerging sex-specific evidence. The group, consisting of seven sessions, foregrounds the menstrual cycle and addresses several ADHD-related topics in relation to this fluctuating pattern: childhood, emotionality, impulsivity, boundaries and safety, and triggers (Figure 1). During the group, participants track their menstrual cycle and possibly related symptoms with a PMS calendar (Appendix A). We advise that this programme be considered in addition to pharmacological and non-pharmacological treatment as usual and not be used as a substitute.

### 4.1. Merging Practice and Research

Several differences in the clinical presentation of ADHD in women and men have been described [4,6,7,24], but controversy exists [3]. As previously described, women with ADHD, when compared to men with ADHD, seem to have more mood [8,9,10], anxiety and eating disorders [9], a higher level of emotional symptoms [11] and relational instability [12]. Social functioning, time perception, and tackling stress also seem more impaired in women with ADHD [10]. The presentation of ADHD in women is described as more complex [11] and no less invalidating [4] than the presentation of ADHD in men. However, women are more often misdiagnosed [23] and diagnosed later [85] than men with ADHD. What is more, female-specific treatment options are scarce. So far, the only female-specific intervention known to us was designed by Gutman et al. [86]. They summarise that women with ADHD struggle with consistency in managing the organisation and stressors of their multiple social roles [86,87]. To target this specific combination of symptoms, they designed an individually tailored intervention for women with (self-reported) ADHD [86]. However, the intervention does not address the menstrual cycle.

During many years of clinical experience, we have come to discern a pattern specific to our female patients. Hormonal fluctuations seem to impact ADHD symptomatology. Available evidence has confirmed this [23,88]. Compared to the general population, women with ADHD are at increased risk of depression and depressive symptoms during periods of ovarian hormone fluctuations [20,41,89,90]. Additionally, during periods of low oestrogen, increased ADHD symptoms [30,41,91,92] and decreased response to psychostimulant medication [25,42,93,94,95] have been described. Moreover, in the periovulatory phase, increased risk-taking behaviour has been reported [19]. High trait impulsivity is central to ADHD and has also been linked to fluctuations in oestrogen and progesterone during the menstrual cycle [96]. Oestrogen is thought to stimulate dopamine synthesis [62,63] and to have a direct and comparable effect on the prefrontal cortex to that of dopamine [34]. Recently, we published a case series that demonstrates the potential benefits of increased psychostimulant dosage in the premenstrual phase. We reported relief of worsening ADHD and mood symptoms premenstrually in all nine consecutive cases [42]. But beyond psychopharmacology, multi-modal treatment for (women with) ADHD is optimal to improve treatment outcome [21]. According to Young et al., non-pharmacological treatment for women with ADHD should include a lifespan approach, giving attention to the complex and developmentally changing presentation of ADHD [23]. Additionally, they and others emphasise that treatment plans should be informed by the interaction between ADHD and fluctuating hormones [19,23,25,26,41]. We agree that the interrelationship between fluctuating sex hormones and ADHD (symptoms) must be addressed in *both* pharmacological and non-pharmacological treatment for women with ADHD [30,62]. Therefore, we have developed a female-specific group programme that foregrounds the effect of cyclical variations in hormones and their influence on mental and physical health.

The core of non-pharmacological treatment for ADHD consists of psychoeducation, support, Cognitive Behavioural Therapy, and practical organisational skills [22]. At PsyQ, all patients are offered a combination of pharmacological and non-pharmacological treatment for ADHD. If indicated, additional interventions are provided. Until we started the group, no female-specific treatment options were available.

### 4.2. A Group Setting

Growing up, girls with ADHD are more likely to have no friends, struggle with social skills and peer interaction, be less popular, and be at greater risk of victimisation [79]. The experience of being a girl with ADHD is covered in the second session of our group programme. Participants considered it an intense but impactful session. Attoe and Climie identified four key themes in their systematic review of the experience of women with an adult diagnosis of ADHD: impacts on social-emotional wellbeing (e.g., low self-esteem, peer relations and emotional control, and coping strategies); difficult relationships; lack of control; and self-acceptance after diagnosis [17]. The group setting, consciously chosen because of the importance of social difficulties for women with ADHD, was valued by the participants as a welcoming and safe space in which they could openly share, even in the presence of strong emotions. Sharing the experience of the group, mutual recognition was greatly appreciated. While the start of the group was stressful for some, participants were impressed by the openness and understanding of other group members. The incorporation of emotions in the group programme offered insight. In adults with ADHD, emotional impulsivity and dysregulation play a substantial role [13,14,15,16]. For women [18], and women with ADHD especially [19,20], the menstrual cycle probably influences these symptoms. For women with ADHD, lack of emotional control adds greatly to their suffering [17]. We deal with these topics in detail in our third and fourth sessions. In ADHD, an interaction between alexithymia, emotion-processing dysfunction and social anxiety has been suggested [97]. Clinically, we consider this a large burden that, if left unaddressed, may impact a woman’s life immensely. Participants of our group found that intense emotional experiences were normalised somewhat, as others had also experienced and described them. In addition, they noted that an increased understanding of the impact of cyclical fluctuations and actively discussing these in the group helped them communicate more confidently and clearly about their cycle and their fluctuating (emotional) needs.

### 4.3. Time and Planning

Inconsistency in regard to reaction time can be considered a stable feature of ADHD, where adults have been described as consistently inconsistent, both behaviourally and in their performance on neurocognitive tests [98]. We argue that some women with ADHD, in the reproductive period of their lives, are *fluctuatingly* inconsistent due to their cyclical nature. Thus, in order to help them manage their symptoms adequately, this additional variability needs to be addressed [42]. In the first session of the group, we introduce the importance of the menstrual cycle. In the third session, we discuss in depth the interrelationship between fluctuating sex hormones and ADHD (symptoms) during different life phases. For the duration of the entire programme, participants track their menstrual cycle and fluctuating ADHD and/or mood symptoms with the PMS calendar (Appendix A). Some women with ADHD feel little control over their lives [17]. Skills training for planning and organising is an essential part of non-pharmacological ADHD treatment [99,100]. We propose that skills training for women with ADHD whose menstrual cycle impacts their mood and cognitive function should explicitly accommodate the menstrual cycle. Thus, ‘realistic planning’ should not only cater to ADHD-specific needs but also help with self-management during the different phases of the cycle (Table 2). The symptoms were similar between individual participants, although not identical, which emphasises the importance of an individually determined, realistic plan.

Initially, we were struck by how unaware women were of (the impact of) their own menstrual cycle. However, ADHD is associated with difficulties in time perception [101,102], so it is not surprising that some women with ADHD have a limited overview of the relationship between the increase in symptoms and the phase of their cycle. In addition, they may struggle to adequately recognise their (fluctuating) wants and needs. Consequently, every month, in the premenstrual phase, women with ADHD may be caught off guard by their worsening ADHD and/or mood symptoms [7,30,40,41]. Similarly, they may be repeatedly surprised mid-cycle by the increase in their risk-taking tendencies [19]. Possibly more poignant, if women do not understand the interaction between fluctuating hormones and ADHD symptoms, they will battle to make sense of their fluctuating course. For this reason, we foreground the menstrual cycle in our programme. In our group, participants became aware of the extent and impact of their own fluctuating symptoms by completing the PMS calendar (Appendix A). For many, the changes associated with menstruation are still shrouded by shame and taboo [103]. Participants in our group emphasised the value of openly sharing with others and experiencing recognition. They recognised themselves in the theory and each other. This, similar to their experience with sharing intense emotions, helped normalise their fluctuating struggles. They began to take their difficulties more seriously. Increased understanding of bodily functions and the interconnectedness with ADHD throughout different life phases allowed for shifts in self-regard and self-acceptance. At evaluation, participants noted that they had incorporated their menstrual cycle in their planning more or had a strong intention to do so in the future. Once they had acknowledged their cyclical pattern, they were able to predict periods of poorer function and adapt their planning in accordance with their cycle. They described their experience as less burdensome.

### 4.4. Masking the Menstrual Cycle

If a professional fails to address the (impact of the) menstrual cycle or ignores the timing aspect of therapeutic interventions, we pose that treatment remains suboptimal at best. Where the menstrual cycle is ignored, women with ADHD may fail to adhere consistently to treatment plans. In ADHD, repeated failure experiences are commonly described [104], and may result in a negative spiral of poor self-esteem, anxiety and depressive symptoms [105,106]. Women, when compared to men with ADHD, appear to have a more pessimistic self-perception [8]. We argue that appropriately tailored treatment plans will minimise experiences of failure and increase adherence to treatment for women with ADHD. Indeed, participants of our programme found the group topics and structure well-tailored to them. During the evaluation, one participant remarked that she had surprised herself, as she had completed her homework almost every time.

A second possible consequence of ignoring the menstrual cycle in ADHD treatment is that women may feel compelled to mask their difficulties, as they may feel embarrassed or not understand the relevance of their ADHD treatment. From an early age, women with ADHD learn to mask their symptoms [58] and adjust their behaviour to the expected social norms [59]. Masking implies continuous confirmation that one needs to adapt to be accepted and, conversely, one is not good enough as herself. In fact, some researchers suggest a relation between this socialisation and the high prevalence of anxiety and depression in girls and women with ADHD [28,87]. ADHD is strongly associated with internalising problems [107], and women with ADHD have more internalising disorders [4] and a more pessimistic self-perception [8] than men. In the second session, we explore the possible consequences of masking from an early age. One participant from the first group explicitly mentioned that the ground rules ‘come as you are’ and ‘do what you need to do to focus best’ were encouraging to minimise masking during the sessions. For the women in our last group, the childhood topic was an ‘impactful eye-opener’. In session two, participants share a photograph of their childhood selves, where they recognise themselves completely, unadjusted to societal expectations. Discussing their girlhood and sharing vulnerabilities made them realise they had masked their ADHD symptoms from childhood onwards. This realisation caused a shift in self-regard, stimulated self-compassion, and made them more aware of what they wanted to change in their lives. Simultaneously, in our experience, the emotional exchange of session two helps stimulate a feeling of group cohesion.

### 4.5. Physically Stressed

Clinically, women with ADHD describe periods when they are more easily overwhelmed, struggle more to maintain an overview, and experience less resilience to stress. Unsurprisingly, these periods often coincide with the premenstrual phase. What is more, in this period, they fail to keep up with obligatory tasks and may have emotional outbursts and/or (severe) depressive or anxious symptoms. This, in turn, augments stress. Stress management was an ongoing topic in our programme. Each session started with a body scan meditation (Figure 1). We included it both as a conscious moment of transition (from work/study/traffic to the group) and as a practice of interoceptive attention. The body scan developed as a part of mindfulness-based stress reduction [108] has been shown to be effective in stress reduction, increased self-compassion, decreased rumination, and medical symptom relief [48]. Between sessions three and four, we introduce positive aspects of ADHD. In session four, we expand on the relation between impulsivity, restlessness and stress and the influence of different phases of the cycle. We also explore ways of harnessing aspects of participants’ own ADHD to break this vicious cycle and reduce stress. The homework of exploring self-care activities between sessions four and five is additionally included to practice stress reduction in different phases of the menstrual cycle. In session five, we then elaborate on the importance of adequate self-management for maintaining or creating (a sense of) safety. Unfortunately, for this particular group, we were forced to amalgamate the fourth session into the fifth. Consequently, there was less focus on rest and relaxation. During the final evaluation, it was suggested that ‘rest and relaxation’ could have been more strongly highlighted in the programme. This again underlines the importance of stress reduction. Furthermore, stress relief seems especially relevant in light of recent findings that connect ADHD, (chronic) stress, and inflammation [109,110].

### 4.6. Understanding and Connection

At evaluation, self-compassion was deemed very important by participants. Recognising and understanding behavioural patterns helped participants have a less judgemental attitude towards their own behaviour. In women who have *fluctuatingly* inconsistent symptoms, we consider physical awareness of great importance. Participants in all three groups described premenstrual binge eating episodes and increased emotional eating. ADHD is associated with disordered eating patterns [111]. Emotional dysregulation and negative affectivity may mediate this association [72], while deficits in interoceptive accuracy also contribute [77]. In community samples, emotional eating appears to be related to the low oestrogen/high progesterone state, amongst other associations [112]. In bulimics, binge frequency increases in the mid-luteal/premenstrual phases [113]. With increased insight concerning premenstrual binge eating episodes, participants were more likely to *care for* their periods of craving by buying slightly less unhealthy but still satisfying foods *in advance*. As they understood their cravings better, they were also more self-forgiving if they ate an entire chocolate bar during their premenstrual period. Some found that this change in attitude from ‘I must not’ to ‘I will try not to, but if it happens now and again, that is alright’ actually resulted in fewer reported binge eating episodes.

*Understanding* and *connection*, the overarching themes distilled from the evaluation, emphasise the importance of offering well-tailored, female-specific information. This helps women with ADHD understand their experience so that they can connect different aspects of their life that they did not know were related. In turn, self-regard and self-management improve, which positively impacts stress and stimulates connection. A deepened understanding of hormonal fluctuations may help women communicate more clearly and confidently about their fluctuating symptoms and have a stronger voice when discussing future (hormonal) treatment options with their own therapist or medical practitioner. Finally, in our group, the experience of being valued and recognised in a real-life setting was appreciated by all participants. Where women with ADHD can share common struggles in an environment that is considered safe, the isolation that stems from masking may be successfully breached.

### 4.7. Clinical Applications

While the initial challenge of organising such a group might seem demanding, we have found several benefits to a group approach. If running the group in its entirety proves unfeasible, we argue that individual implementation of (parts of) this programme may be worthwhile nonetheless. For example, recognition of the relationship between hormonal fluctuations and emotional and cognitive states could be included in the psychological management of women with ADHD. Amongst patients and practitioners alike, there is a great demand for female-specific treatment options for ADHD. While the number of participants in the first three groups is small, some of the participants have gone on to educate those around them directly and/or via social media. In addition, a significant number of our participants were mothers and intended to use the acquired knowledge and skills to educate their (female) children (with ADHD).

### 4.8. Limitations and Strengths

While this is the first time, to our knowledge, that a group programme for women with ADHD is presented taking the menstrual cycle into account, our project does have some limitations. Firstly, although we describe the group programme in great detail, we have analysed the evaluation of only one group (the last). The experience of the first two groups is not included in the current presentation. However, they did provide positive feedback and considered the group a valuable addition to treatment as usual. By presenting this group programme, which is subject to continuous development, we hope to inspire others to meet this unmet need and treat women with ADHD as *women*. A second potential limitation is that the final evaluation of the group, which was relatively short, was conducted by MdJ and DW, who also designed and led the group. Although we explicitly emphasised the importance of full openness and honesty, this might have hindered participants from expressing negative feedback. In the evaluation, the participants explicitly denied that they felt constrained, and both positive and negative feedback was shared. Additionally, to ensure the rigour of our qualitative analysis, we used triangulation of researchers, wherein MM, who had no relation with the programme or participants whatsoever, functioned as an independent researcher. We also consider it a limitation that we have so far only completed three groups. However, when reflecting back on the first two groups and our experience with the third, we do feel like the current presentation is an accurate representation. Strikingly, no participants in the third group had a co-occurring diagnosis of PMDD at the start of the programme. After completing the PMS calendar for several months, participants gained insight into their fluctuating symptoms and developed a stronger voice in discussing possible adjustments to their diagnoses and further treatment options with their individual therapists. Finally, while the (subjective) impact of the group was discussed during the evaluation, we did not measure the effect of the group. Nonetheless, we hope to provide a starting point for a female-specific treatment perspective for women with ADHD.

### 4.9. Future Directions

Firstly, it is important to objectivate the effect of our group programme by including standardised before- and after measurements. Including a longer follow-up may also be helpful. In the future, we will no longer include participants who are at the very beginning of their ADHD treatment. This will ensure optimal group dynamics, less discrepancy between group members, and a better use of group time. We advise other practitioners to adopt this approach, too. The field would benefit from the implementation of our programme by others and the evaluation of their (patients’) experiences. Beyond the menstrual cycle, we aim to create similar specific groups for perimenopausal and post-natal women with ADHD, as these are other periods in women’s reproductive lives that are characterised by hormonal fluctuations that impact mood and ADHD symptoms. Additionally, it would be interesting to implement a similar group programme for women with other psychiatric conditions, as ADHD is not the only disorder that is associated with cyclically fluctuating symptoms (e.g., autism spectrum disorder [114], schizophrenia [115], and bipolar disorder [116]) [18]. We are examining ways to address the expressed wish of participants for a follow-up group. Finally, we are considering constructive ways of including the (life) partners of group members into the programme.

## 5. Conclusions

Fluctuating sex hormones appear to influence ADHD symptomatology. Therefore, the menstrual cycle needs to be taken into account in both pharmacological and non-pharmacological treatment for women with ADHD. We have described this first group programme specifically for women with ADHD and self-reported premenstrual worsening of ADHD and/or mood symptoms and qualitatively analysed the final evaluation of our last group. Female-specific psychoeducation combined with a tailored treatment plan may enhance adherence and clinical outcomes for these women. By taking the menstrual cycle into account in ADHD, the clinician avoids reinforcing maladaptive behaviour and/or compounding experiences of failure. Our report will hopefully open the door to improving awareness of the hormonal impact on ADHD (treatment) for women with ADHD and all who treat them.

## Figures and Tables

**Figure 1 jcm-13-02106-f001:**
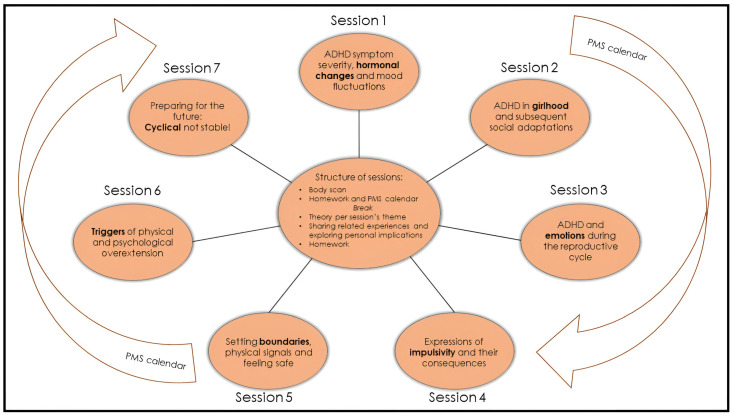
Overview of the female-specific treatment group for ADHD. The programme consists of 7 sessions of two hours that are given every other week, and all have their own focus. The PMS calendar functions as a connecting thread (Appendix A). All sessions have a similar structure. Every session starts with a 5 min body scan meditation. Homework and the PMS calendar are discussed. After a break, the content section consists of a combination of theory and (individual) experiences. Every session ends with the homework assignment for the next session. During the body scan, under verbal instruction from a therapist, participants are encouraged to move attention sequentially via the body, paying attention to present moment sensory experience in each body area without trying to change anything [48]. The PMS calendar was devised and adjusted in collaboration with participants of previous groups. For the duration of the group programme (about 16 weeks), participants use it to track their menstrual cycle, ADHD, mood, and somatic symptoms.

**Figure 2 jcm-13-02106-f002:**
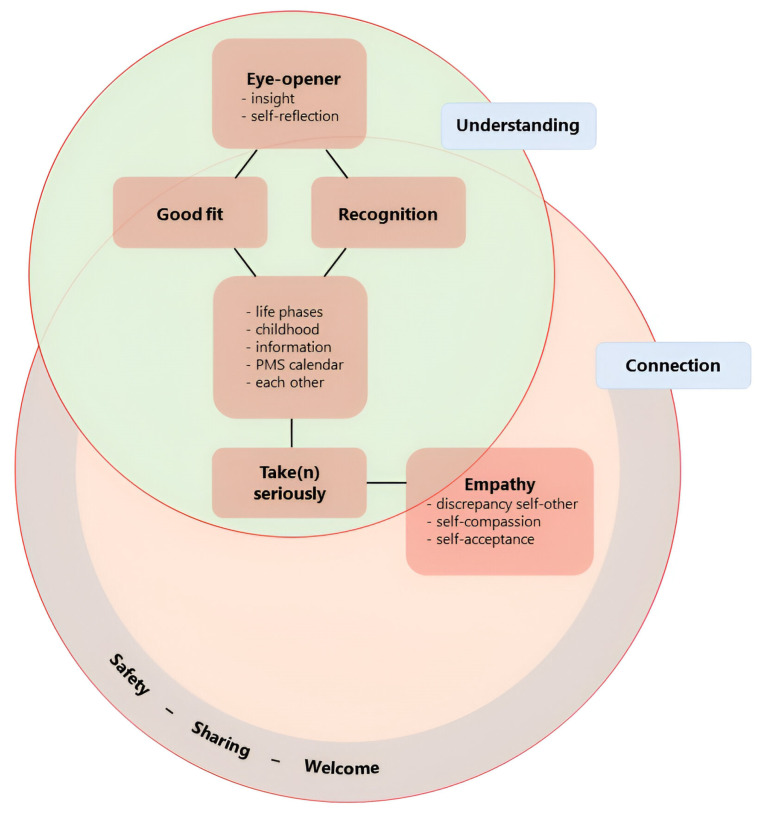
Graphical summary of themes from final evaluation and their interrelationship. Six key themes were identified via our qualitative analysis of the final evaluation: eyeopener, good-fit, recognition, take(n) seriously, empathy, and safety-sharing-welcome. We decided on a compound theme for safety-sharing-welcome as we felt that the separate meanings were similar enough to group but not identical enough to merge. All key themes are literal (translations of) expressions of our participants. The overarching themes, which overlap, are understanding and connection.

**Table 1 jcm-13-02106-t001:** Participant characteristics.

	Mean (Range)
Age *	37.8 years (29–46 years)
Time since ADHD diagnosis	32 months (4 months–10 years)
Time since treatment start **	10 months (2–22 months)
	n⁰ (total = 6)
Use of ADHD medication at start *	4
Stable medication *	2
Use of ADHD medication at the end ***	5
Stable medication ***	3

* At the start of the group programme ** Time since the commencement of treatment for ADHD at PsyQ, The Hague *** At the end of the group programme.

**Table 2 jcm-13-02106-t002:** Realistic planning per menstrual phase.

	Follicular Phase	Periovulatory Phase	Premenstrual Phase
Description	Approximately the first two weeks of the menstrual cycle. From day 1 of menses until ovulation. Term used loosely: from the moment during menstruation when subjective premenstrual symptoms start improving until the periovulatory phase.	The days around ovulation. In our groups, the exact moment of ovulation is unknown. We rely on subjective reports of ‘ovulation-like’ symptoms. Importantly, some women experience mild ‘premenstrual symptoms’ directly after ovulation.	This is the period before menstruation commences. This phase is also known as ‘week 4′ of the menstrual cycle and is characterised by lower oestrogen and progesterone levels.
Symptoms	More energy and focus	Increased risk-taking behaviour	Worse mood and ADHD symptoms
Plan for	Particularly challenging/boring tasks	Sufficiently stimulating activities	Increased periods of ‘down-time’
Examples	Job interviews, house cleaning, buying groceries in bulk for other phases, completing administration, preparing for big deadlines Plan ahead (realistically) Discuss what is needed during premenstrual phase with (life)partners	Engaging in hobbies, such as cleaning out an old wardrobe Socialising Avoid ‘dangerous’ situations (e.g., clubbing)	Simple and enjoyable tasks that have been ‘saved up’ Fewer social obligations Avoid big decisions Relaxing activity (e.g., TV series, massage)

Note: the suggested examples and plans are not exhaustive or ideal.

## Data Availability

The datasets presented in this article are not readily available because we are describing clinical findings, and patient privacy will be protected.

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
