# Peer review of "A Female-Specific Treatment Group for ADHD—Description of the Programme and Qualitative Analysis of First Experiences"

_jcm, 2024, doi:10.3390/jcm13072106_

Round 1

Reviewer 1 Report

Comments and Suggestions for Authors

The manuscript examines an often overlooked aspect of treatment of ADHD in women and the poor attention effects of hormonal variations on ADHD symptoms receive. The group intervention described is very unique and innovative, combining several aspects of psychoeducation, feedback, cognitive restructuring etc. 

However, it is hard to make a case for the inclusion of such an intervention without adequate data on the benefit of this intervention. While the qualitative analysis is helpful in understanding the subjective experiences of patients with ADHD, it would be interesting to see the potential benefit patients derive from this intervention since it is a time and resource intensive plan with less scope for large-scale applicability. Adding some insights into the benefit/efficacy of this intervention may interest practitioners more and incentivize them to add the same to their management plan. 

The manuscript needs to be considerably edited to make it more concise - especially the introduction section. 

Comments on the Quality of English Language

The MS is verbose and has punctuation errors. Please edit to make it more comrephensible. 

Author Response

Thank you for your rapid and helpful review! Please see the attachment.

Reviewer 2 Report

Comments and Suggestions for Authors

Thank you very much for the invitation to perform a peer review of the research report “A Female-Specific Treatment Group for ADHD – Description of the Programme and Qualitative Analysis of First Experiences”. 

Overview of the findings

In essence, the paper provides a rich (if relatively ‘humble’ in terms of the sample size) and novel body of evidence on the effectiveness of the original, multidimensional, and—first and foremost—women-focused treatment strategy for attention-deficit / hyperactivity disorder (ADHD). The good news is that it just seems to work well.

Notably, the authors both devised and subsequently tested the above-mentioned strategy within the clinical realm. This feature of the research project deserves to be appreciated separately, since the level of intellectual and ‘executive’ independence I have just referred to (i.e., all the ‘steps’ involved in the scientific process, from delineating the concept of the study, all the way to the clinical testing) is quite a rarity in contemporary academia.

Originality 

As already mentioned, the project clearly represents a novel yet impeccably logical approach to the specific gender-related challenges of ADHD among women. What I mean by the study’s ‘impeccable logic’ is that nothing came out of the blue in this case. The authors made pieces of the jigsaw fall into the right places, eventually translating into a piece of more than just good science. Hence, it is rather needless to say (at this stage of the peer review!) that the study is vastly original – and potentially highly consequential as well (given the fact that the implications of the research project are nearly instantly applicable to the day-to-day clinical practice worldwide).

Importance of the work to general readers

Bearing in mind that ADHD affects approximately 1.5–5% of the general population (Fayyad et al. 2017; Kooij et al. 2019; Polanczyk et al. 2015), it would be rather difficult to overemphasize the importance of the study to a vast number of non-clinicians (obviously, as well as to medical professionals).

Scientific reliability

From the methodological standpoint, finding any major weaknesses in the study would be next to impossible. For this reason, scrutinizing the paper for any minor (i.e., irrelevant) ‘lapses’ would be just nasty. To sum things up, I have no suggestions for additional refinements.

References

Fayyad J, Sampson NA, Hwang I, Adamowski T, Aguilar-Gaxiola S, Al-Hamzawi A, Andrade LH, Borges G, de Girolamo G, Florescu S, Gureje O, Haro JM, Hu C, Karam EG, Lee S, Navarro-Mateu F, O'Neill S, Pennell BE, Piazza M, Posada-Villa J, Ten Have M, Torres Y, Xavier M, Zaslavsky AM, Kessler RC, Collaborators WHOWMHS (2017) The descriptive epidemiology of DSM-IV Adult ADHD in the World Health Organization World Mental Health Surveys. Atten Defic Hyperact Disord 9: 47-65.

Kooij JJS, Bijlenga D, Salerno L, Jaeschke R, Bitter I, Balazs J, Thome J, Dom G, Kasper S, Nunes Filipe C, Stes S, Mohr P, Leppamaki S, Casas M, Bobes J, McCarthy JM, Richarte V, Kjems Philipsen A, Pehlivanidis A, Niemela A, Styr B, Semerci B, Bolea-Alamanac B, Edvinsson D, Baeyens D, Wynchank D, Sobanski E, Philipsen A, McNicholas F, Caci H, Mihailescu I, Manor I, Dobrescu I, Saito T, Krause J, Fayyad J, Ramos-Quiroga JA, Foeken K, Rad F, Adamou M, Ohlmeier M, Fitzgerald M, Gill M, Lensing M, Motavalli Mukaddes N, Brudkiewicz P, Gustafsson P, Tani P, Oswald P, Carpentier PJ, De Rossi P, Delorme R, Markovska Simoska S, Pallanti S, Young S, Bejerot S, Lehtonen T, Kustow J, Muller-Sedgwick U, Hirvikoski T, Pironti V, Ginsberg Y, Felegyhazy Z, Garcia-Portilla MP, Asherson P (2019) Updated European Consensus Statement on diagnosis and treatment of adult ADHD. Eur Psychiatry 56: 14-34.

Polanczyk GV, Salum GA, Sugaya LS, Caye A, Rohde LA (2015) Annual Research Review: A meta-analysis of the worldwide prevalence of mental disorders in children and adolescents. J Child Psychol Psychiatry 56: 345-65.

Comments on the Quality of English Language

Some minor 'lapses' could have been spotted in the manuscript. However, given the fact that I am not a native English speaker, the above-mentioned impression of mine might be just wrong. 

Author Response

Thank you for your rapid and positieve review! Please see the attachment.

Round 2

Reviewer 1 Report

Comments and Suggestions for Authors

Thank you for revising the manuscript and providing clarifications.